# Alcohol consumption in the general population is associated with structural changes in multiple organ systems

Evangelos Evangelou[1,2], Hideaki Suzuki[3,4,5†], Wenjia Bai[5,6†], Raha Pazoki[7,8], He Gao[1,7], Paul M Matthews[5,9,10], Paul Elliott[1,7,9,10,11]*

[1]Department of Epidemiology and Biostatistics, School of Public Health, Imperial College London, London, United Kingdom; [2]Department of Hygiene and Epidemiology, University of Ioannina Medical School, Ioannina, Greece; [3]Department of Cardiovascular Medicine, Tohoku University Hospital, Sendai, Japan; [4]Tohoku Medical Megabank Organization, Tohoku University, Sendai, Japan; [5]Department of Brain Sciences, Imperial College London, London, United Kingdom; [6]Data Science Institute, Imperial College London, London, United Kingdom; [7]MRC Centre for Environment and Health, School of Public Health, Imperial College London, London, United Kingdom; [8]Division of Biomedical Sciences, Department of Life Sciences, College of Health, Medicine and Life Sciences, Brunel University London, London, United Kingdom; [9]UK Dementia Research Institute at Imperial College London, London, United Kingdom; [10]National Institute for Health Research Imperial College Biomedical Research Centre, Imperial College London, London, United Kingdom; [11]British Heart Foundation Centre for Research Excellence, Imperial College London, London, United Kingdom

*For correspondence:
p.elliott@imperial.ac.uk

†These authors contributed equally to this work

## Abstract

**Background:** Excessive alcohol consumption is associated with damage to various organs, but its multi-organ effects have not been characterised across the usual range of alcohol drinking in a large general population sample.

**Methods:** We assessed global effect sizes of alcohol consumption on quantitative magnetic resonance imaging phenotypic measures of the brain, heart, aorta, and liver of UK Biobank participants who reported drinking alcohol.

**Results:** We found a monotonic association of higher alcohol consumption with lower normalised brain volume across the range of alcohol intakes ($-1.7 \times 10^{-3} \pm 0.76 \times 10^{-3}$ per doubling of alcohol consumption, p=$3.0 \times 10^{-14}$). Alcohol consumption was also associated directly with measures of left ventricular mass index and left ventricular and atrial volume indices. Liver fat increased by a mean of 0.15% per doubling of alcohol consumption.

**Conclusions:** Our results imply that there is not a 'safe threshold' below which there are no toxic effects of alcohol. Current public health guidelines concerning alcohol consumption may need to be revisited.

**Funding:** See acknowledgements.

## Introduction

Alcohol consumption causes damage to multiple organs and systems, and heavy drinking is associated with increased all-cause mortality (*Bell et al., 2017*). According to the Global Burden of Diseases Study, alcohol use was the seventh leading risk factor for both deaths and disability-adjusted

life years in 2016, accounting for 2.2% and 6.8% excess in age-standardized female and male deaths, respectively (*GBD 2016 Alcohol Collaborators, 2016*). While previous evidence has suggested that low to moderate amounts of daily consumption may have beneficial effects on cardiovascular health (*Bell et al., 2017*), a recent large-scale meta-analysis concluded that even moderate daily alcohol intake may have significant impact on disease risk (*Wood et al., 2018*). Because of these uncertainties, there remains controversy about whether there is a 'safe level' of alcohol drinking for the general population (*Fernández-Solà, 2015*; *Mukamal and Rimm, 2008*).

The liver is a primary target for the detrimental effects of alcohol, as it is the primary site of alcohol metabolism (*Cederbaum, 2012*). With high levels of alcohol consumption, effects on other organs (including the brain and heart) have been described (*Obad et al., 2018*). Excessive alcohol use during adolescence has been associated with reduced brain grey matter volumes (*Heikkinen et al., 2017*), but evidence regarding structural brain changes at lower levels of alcohol intake is limited and conflicting (*Ding et al., 2004*; *Mukamal et al., 2001*; *McEvoy et al., 2018*). Moderate to heavy alcohol consumption is implicated causally with pathologically reduced left ventricular ejection fraction (*van Oort et al., 2020*), cardiomyopathy, heart failure, and sudden death. Analyses of cardiac structure based on echocardiography have suggested that smaller differences in left ventricular mass consistent with early pathology can also be attributed to lower levels of alcohol intake (*Voskoboinik et al., 2019*; *Gonçalves et al., 2015a*; *Gémes et al., 2018*).

Here, for the first time, we report associations across the range of population alcohol consumption with differences in morphology or function of multiple organs determined from quantitative measures of the brain, cardiac structure and function, and liver fat magnetic resonance imaging (MRI) scans. Our aim was to investigate effects of alcohol at intakes within the currently recommended limits for consumption by the general population. Discovery of evidence for potentially toxic effects of alcohol within these recommended limits would have important implications for public health and government policies regarding 'safe' levels of alcohol drinking.

## Materials and methods

### Study participants

UK Biobank is a prospective, observational study of ~500,000 people across the United Kingdom, aged 40–69 years at recruitment (2007–2010) (*Sudlow et al., 2015*; *Bycroft et al., 2018*). Here we used a subset of the UK Biobank data from participants whose brain, cardiac and/or aortic, and liver MRI images and image-derived phenotypes (IDPs) were available. Non-drinkers and those with self-reported brain, cardiac, and/or aortic diseases were excluded. IDPs of participants were included based on availability of measures after the application of exclusion criteria (brain grey matter [N = 10,143], brain white matter [N = 9053], heart [N = 11,821], aortic [N = 12,376], and liver [N = 3649]) (*Figure 1*). *Table 1* describes characteristics of the population included in the analyses. The study is reported following the Strengthening the Reporting of Observational Studies in Epidemiology (STROBE) guideline.

### Baseline characteristics

Information on age, sex, ethnicity, college degree education, body mass index (BMI), hypertension, diabetes, and history of smoking and cardiac, brain, and/or aortic diseases were reported at the imaging assessment. We defined participants as hypertensive if they had systolic blood pressure $\geq$ 140 mmHg or diastolic blood pressure $\geq$ 90 mmHg or were receiving antihypertensive medication (*Suzuki et al., 2017*). We recorded self-reported diabetes, smoking history, and college degree education.

### Alcohol consumption

We calculated alcohol intake as grams of alcohol per day (g/d) among drinkers based on self-reported alcohol drinking from a touch-screen questionnaire described previously (*Evangelou et al., 2019*). Briefly, the quantity of each type of drink was multiplied by its standard drink size and reference alcohol content. Drink-specific intake during the reported drinking period was summed and converted to g/d alcohol intake for each participant with completed responses to the quantitative drinking questions. The alcohol intake for participants with incomplete responses was imputed by

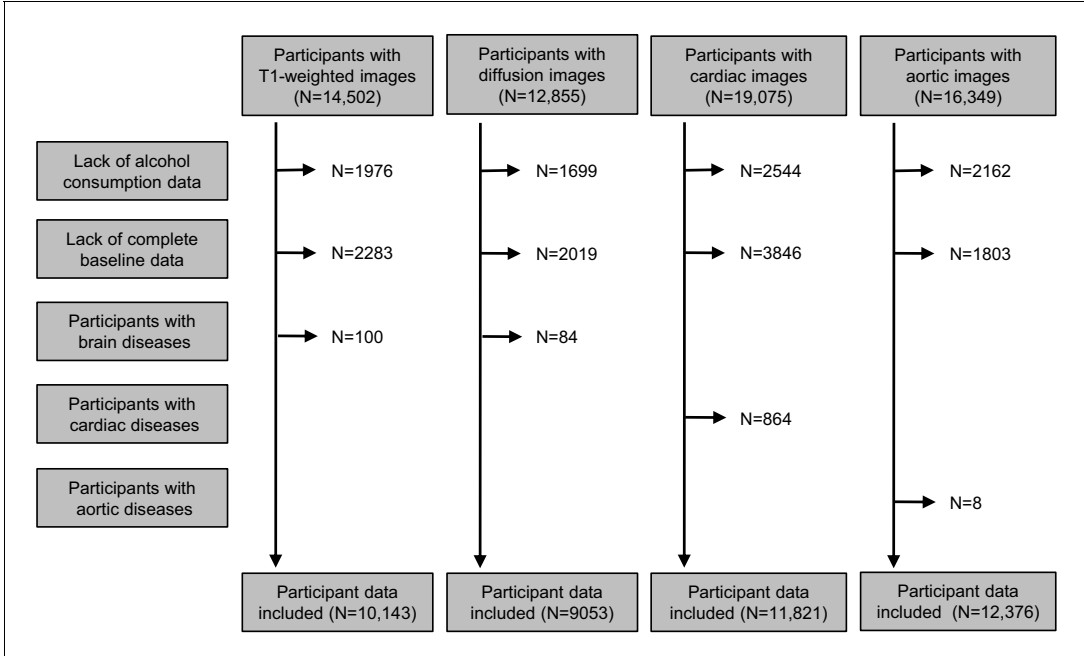

**Figure 1.** Flow chart of eligible participants included in analyses.

bootstrap resampling from the completed responses, stratified by drinking frequency and sex. Alcohol intake was $\log_2$-transformed, as it has a skewed distribution. Using this transformation, a $\log_2$ change of 1 unit translates to a doubling of alcohol consumption, e.g., from 10 g/d to 20 g/d.

## Brain MRI acquisition and pre-processing

Details of the image acquisition are available online (*Miller et al., 2016*). Briefly, the T1-weighted (3D MPRAGE, $1 \times 1 \times 1$ mm³ resolution, field of view [FOV]/matrix = $208 \times 256 \times 256$, TR [repetition time] = 2000 ms, TI [inversion time] = 880 ms) brain images used here were acquired using a Siemens Skyra 3T running VD13A SP4 (Siemens Healthcare, Erlangen, Germany) with a Siemens 32-channel RF receive head coil used for structural analyses. Before analyses, the images were registered in the standard Montreal Neurological Institute (MNI) space using DARTEL tools in SPM12 (https://www.fil.ion.ucl.ac.uk/spm/software/spm12/).

**Table 1.** Participant characteristics.

| | Brain grey matter | Brain white matter | Heart | Aorta | Liver |
|---|---|---|---|---|---|
| N | 10,143 | 9053 | 11,821 | 12,376 | 3649 |
| Baseline characteristics | | | | | |
| Age (years) (mean ± SD) | 62.9 ± 7.4 | 62.9 ± 7.4 | 62.8 ± 7.4 | 63.0 ± 7.4 | 55.7 ± 7.5 |
| Male (%) | 49.8 | 49.4 | 49.9 | 50.2 | 50.5 |
| Caucasian (%) | 99.8 | 99.8 | 99.8 | 99.8 | 93.2 |
| Educational attainment (%) | 53.2 | 53.5 | 53.5 | 52.9 | 53.1 |
| Body mass index (mean ± SD) | 26.7 ± 4.4 | 26.6 ± 4.3 | 26.5 ± 4.3 | 26.5 ± 4.2 | 26.5 ± 4.1 |
| Hypertension (%) | 39.6 | 39.3 | 39.0 | 39.1 | 46.5 |
| Diabetes (%) | 5.2 | 5.0 | 4.7 | 4.9 | 2.5 |
| Smoking history (%) | 39.7 | 39.4 | 38.6 | 39.1 | 36 |
| Alcohol consumption (g/d) (median–IQR) | 14.29 (6.46–26.78) | 14.29 (6.26–26.79) | 14.29 (6.70–26.79) | 14.29 (6.69–26.79) | 16.61 (8.93–28.86) |

After separate segmentation of grey and white matter and cerebrospinal fluid, each tissue mask was modulated with the Jacobian determinants derived from the spatial normalisation, multiplying each voxel by the relative change in volume to correct for volume changes in the non-linear normalisation (*Good et al., 2001*). Brain and regional volumes were normalised to the corresponding total intracranial volumes, calculated from the sums of volumes of the grey and white matter and cerebrospinal fluid. For the multiple regression voxel-wise analysis, the normalised grey matter maps were smoothed by convolving an isotropic Gaussian kernel of 8 mm full width at half maximum, excluding voxels with a grey matter probability value < 0.2.

Brain diffusion MRI images were acquired using a Stejskal-Tanner pulse sequence (*Elliott et al., 2018*). Our analyses used the white matter microstructural IDPs for fractional anisotropy and orientation dispersion (the extent of directional complexity of diffusion) (*Zhang et al., 2012*; *Wood et al., 2018*) for 27 probabilistically defined (*Suzuki et al., 2017*) white matter tracts described and made available in the UK Biobank Data Showcase (*UK Biobank, 2021*). The white matter microstructure measures were then expressed as mean z-scores (referenced to the mean values for the full study population) in our analyses.

## Cardiac and aortic MRI acquisition and pre-processing

Details of the cardiac and aortic image acquisitions were reported previously (*Petersen et al., 2016*). Briefly, the cardiac and aortic MRI were acquired using a clinical wide bore 1.5T scanner (MAGNETOM Aera, Syngo Platform VD13A, Siemens Healthcare, Erlangen, Germany) with 48 receiver channels, a 45 mT/m and 200 T/m/s gradient system, and an 18-channel anterior body surface coil used in combination with 12 elements of an integrated 32 element spine coil and electro-cardiographic gating for synchronisation with the cardiac cycle. The acquired images were segmented to derive IDPs using a fully convolutional network (CNN) (*Bai et al., 2018*; *Wenjia Bai et al., 2018*).

The ventricular CNN image segmentation provided measures that, with adjustments for body surface area, were used as IDPs for the left ventricular mass, left ventricular end-diastolic (LVEDVI) and left ventricular end-systolic volume (LVESVI), and right ventricular end-diastolic (RVEDVI) and right ventricular end-systolic volume (RVESVI) indices. Left and right ventricular ejection fraction IDPs were derived from integrations of the primary indices as (LVEDVI – LVESVI)/LVEDVI $\times$ 100 and (RVEDVI – RVESVI)/RVEDVI $\times$ 100, respectively. The atrial image segmentation provided left and right atrial volume indices after adjustment for body surface area. The aortic image segmentation provided maximal ascending (AAoAI) and descending (DAoAI) aortic area indices and minimal ascending ($AAoAI_{min}$) and descending ($DAoAI_{min}$) aortic area indices after adjustment for body surface area. Ascending and descending aortic distensibilities were derived as (AAoAI – $AAoAI_{min}$)/$AAoAI_{min}$/(systolic – diastolic blood pressure) and (DAoAI – $DAoAI_{min}$)/$AAoAI_{min}$/(systolic – diastolic blood pressure), respectively (*Petersen et al., 2016*).

## Liver fat MRI acquisition and pre-processing

Abdominal images for assessments of liver fat were acquired using a Siemens 1.5T MAGNETOM Aera. Details of the MRI acquisition and pre-processing protocol are provided elsewhere (*Wilman et al., 2017*). Briefly, a dual-echo Dixon Vibe protocol, which can be used to generate images distinguishing water and fat, from which liver fat could be determined, was performed ($2.2 \times 1.2 \times 10$ mm$^3$ resolution, TR = 3.23 ms, TE = 1.44 ms). The liver MRI proton density fat fraction % derived is available to researchers through the UK Biobank Data Showcase (*UK Biobank, 2021*).

## Statistical analysis

We estimated the age-related differences in the brain normalised volume, cardiac, and liver fat IDPs by their regression onto age adjusted for sex, ethnicity, educational level, BMI, hypertension, diabetes, and smoking history. We then examined the magnitudes of differences in organ morphology or functional IDPs with alcohol consumption. Each IDP was regressed onto log$_2$-transformed alcohol consumption adjusted for age, sex, ethnicity, educational level, BMI, hypertension, diabetes, and smoking history; raw coefficients are used for all measures except brain white matter diffusion measures, for which standardised coefficients are reported. Normality of the IDPs was tested using a

Shapiro–Wilks test. We used partial residual plots to assess any deviation from the fitted model. Additionally, a voxel-wise parametric analysis (*Fernández-Solà, 2015*; *McEvoy et al., 2018*), which used each voxel of the grey matter maps as dependent variable and $\log_2$-transformed alcohol consumption as independent variable adjusted for intracranial volume and the same covariates as above, was conducted for mapping grey matter regions associated with alcohol consumption. In secondary analysis of brain, cardiac, and liver phenotypes, an interaction term for age and $\log_2$ alcohol consumption was included in the regression models. To correct for multiple comparisons, the significance level was set to $p<0.017$ and $p<4.5 \times 10^{-3}$ for brain and heart/aorta imaging IDPs, respectively, whereas for liver fat at $p<0.05$. For the voxel-wise analysis, a family-wise error-corrected threshold of $p<0.05$ was used for grey matter analysis. All statistical analyses were carried out using STATA 14.

### Patient involvement

The performed analyses are based on existing data from a population-based cohort in the United Kingdom (UK Biobank). No patients were explicitly engaged in designing the present research question or the outcome measures, nor were they involved in developing plans for recruitment, design, or implementation of the study. No patients were asked to advise on interpretation or writing up of results. Results from UK Biobank are routinely disseminated to study participants via the study website and social media outlets.

## Results

### Participant characteristics and imaging phenotypes

Baseline characteristics of the participants included in this study and summary IDPs are shown in *Tables 1* and *2*. For the five subsets in our analysis, median alcohol intakes among these drinkers were similar: ~20.9 g/d (i.e., just over two 10 g drinking units, where 125 ml of 12.5% wine is 1.25 drinking units) for men and ~10.7 g/d for women with 25th and 75th centiles ~10.3 g/d and 35.8 g/d for men and ~3.6 g/d and 17.9 g/d for women (*Supplementary file 1*).

**Table 2.** Structural imaging phenotypes for brain (N = 10,143), heart (N = 11,821), and aorta (N = 12,376) in the UK Biobank.

| Imaging-derived phenotypes | Mean ± SD |
|---|---|
| Brain | |
| Normalised brain volume | 0.72 ± 0.045 |
| Normalised grey matter volume | 0.43 ± 0.034 |
| Normalised white matter volume | 0.29 ± 0.020 |
| Heart | |
| Left ventricular mass index (g/m$^2$) | 46.2 ± 8.5 |
| Left ventricular end-diastolic volume index (ml/m$^2$) | 80.0 ± 13.7 |
| Left ventricular ejection fraction (%) | 59.6 ± 5.9 |
| Left atrial volume index (ml/m$^2$) | 38.8 ± 10.7 |
| Right ventricular end-diastolic volume index (ml/m$^2$) | 84.5 ± 15.4 |
| Right ventricular ejection fraction (%) | 57.3 ± 6.0 |
| Right atrial volume index (ml/m$^2$) | 46.1 ± 13.1 |
| Aorta | |
| Ascending aortic area index (mm$^2$/m$^2$) | 455.3 ± 91.9 |
| Ascending aortic distensibility ($10^{-3}$mmHg$^{-1}$) | 1.98 ± 1.17 |
| Descending aortic area index (mm$^2$/m$^2$) | 254.1 ± 43.3 |
| Descending aortic distensibility ($10^{-3}$mmHg$^{-1}$) | 2.64 ± 1.25 |

SD: standard deviation.

## Associations of alcohol consumption with brain structure

Age-related differences in normalised brain volumes (NBV) in the population, adjusted for alcohol consumption, were about 0.3% lower/year (mean ± standard error: $-3.0 \times 10^{-3} \pm 0.05 \times 10^{-5}$/year, $p<1.0 \times 10^{-300}$), consistent with previous studies (*Enzinger et al., 2005*). The contribution of alcohol to the observed brain volume differences was about 0.17% lower NBV per doubling of alcohol consumption ($-1.7 \times 10^{-3} \pm 2.3 \times 10^{-4}$, $p=3.0 \times 10^{-14}$). Lower volumes per doubling of alcohol consumption of both total grey ($-1.2 \times 10^{-3} \pm 1.7 \times 10^{-4}$, $p=1.9 \times 10^{-12}$) and white ($-5.1 \times 10^{-4} \pm 1.2 \times 10^{-4}$, $p=2.1 \times 10^{-5}$) matter jointly account for the lower brain volumes associated with greater alcohol intake. Partial residual plots confirmed the observed relationship without any deviations from the fitted model (*Figure 2a–c*).

Exploration of voxel-wise parametric associations of the $\log_2$-transformed g/d alcohol intake with brain grey matter showed greatest negative associations with regions in the cingulate and orbital frontal cortices, the bilateral insula, and thalami (*Figure 3*). There were no positive correlations between alcohol and grey matter volumes for any of the brain regions after family-wise error correction.

Finally, given the associations of greater alcohol intake with lower white matter volumes, we explored alcohol-associated differences in fractional anisotropy, a measure of white matter microstructure, across 27 major white matter tract IDPs. The bilateral corticospinal tracts showed increased fractional anisotropy with greater alcohol intake (standardised coefficient, left, $0.013 \pm 0.003$ per doubling of alcohol consumption, $p=1.0 \times 10^{-4}$; right, $0.011 \pm 0.003$, $p=4.20 \times 10^{-4}$). This was associated with lower orientation dispersion (standardised coefficient, left, $-0.013 \pm 0.003$, $p=8.1 \times 10^{-4}$; right, $-0.013 \pm 0.003$, $p=1.4 \times 10^{-3}$), suggesting greater fibre coherence or a relatively reduced density of orthogonally crossing white matter tracts (*Zhang et al., 2012*; *Mollink et al., 2017*).

## Associations of alcohol consumption with heart and aorta

We first tested for age-related differences in cardiac and aortic IDPs adjusted for alcohol consumption in the population. Left ventricular mass index and the left atrial and left and right ventricular end diastolic volume indices were lower with greater age. Right and left ventricular ejection fractions both were modestly greater with age. There was also a small relative increase in the right atrial volume index with age (*Table 3*).

$\log_2$-transformed alcohol consumption was associated directly with measures of left ventricular and atrial mass and volume. The effects of alcohol on the cardiac IDPs were largely in opposite direction to those for age (*Table 3*).

Ascending and descending aortic area indices increased with age, while distensibility decreased; $\log_2$-transformed alcohol consumption associations for the aorta showed the same directions of effect as age (*Table 3*).

The associations with alcohol appeared linear-log with no deviation as indicated by the partial residual plots (*Figure 2d–n*). We also explored interactions between age and $\log_2$ alcohol consumption, which were most evident for the left ventricular mass index and aortic distensibility IDPs (*Supplementary file 2*). There was no evidence for U-shaped associations, i.e., higher values at both the lower and higher ends of the alcohol intake distribution, for any of the aortic or cardiac IDPs.

## Association of alcohol consumption with liver fat

We did not observe age-related differences in liver fat after adjusting for alcohol consumption and other relevant variables ($0.015 \pm 0.01$, $p=0.14$). There was an increase of liver fat per doubling of alcohol consumption ($0.15 \pm 0.06$, $p=0.006$), with no deviation from linear-log association observed (*Figure 2o*).

## Discussion

In this large population-based study of the effects of alcohol consumption on different organs, we found that increasing alcohol intake was associated with reduced brain grey matter volume, increased left ventricular mass and volume and aortic area index, and reduced descending aortic distensibility and increased liver fat. There was no evidence against a monotonic increase across the

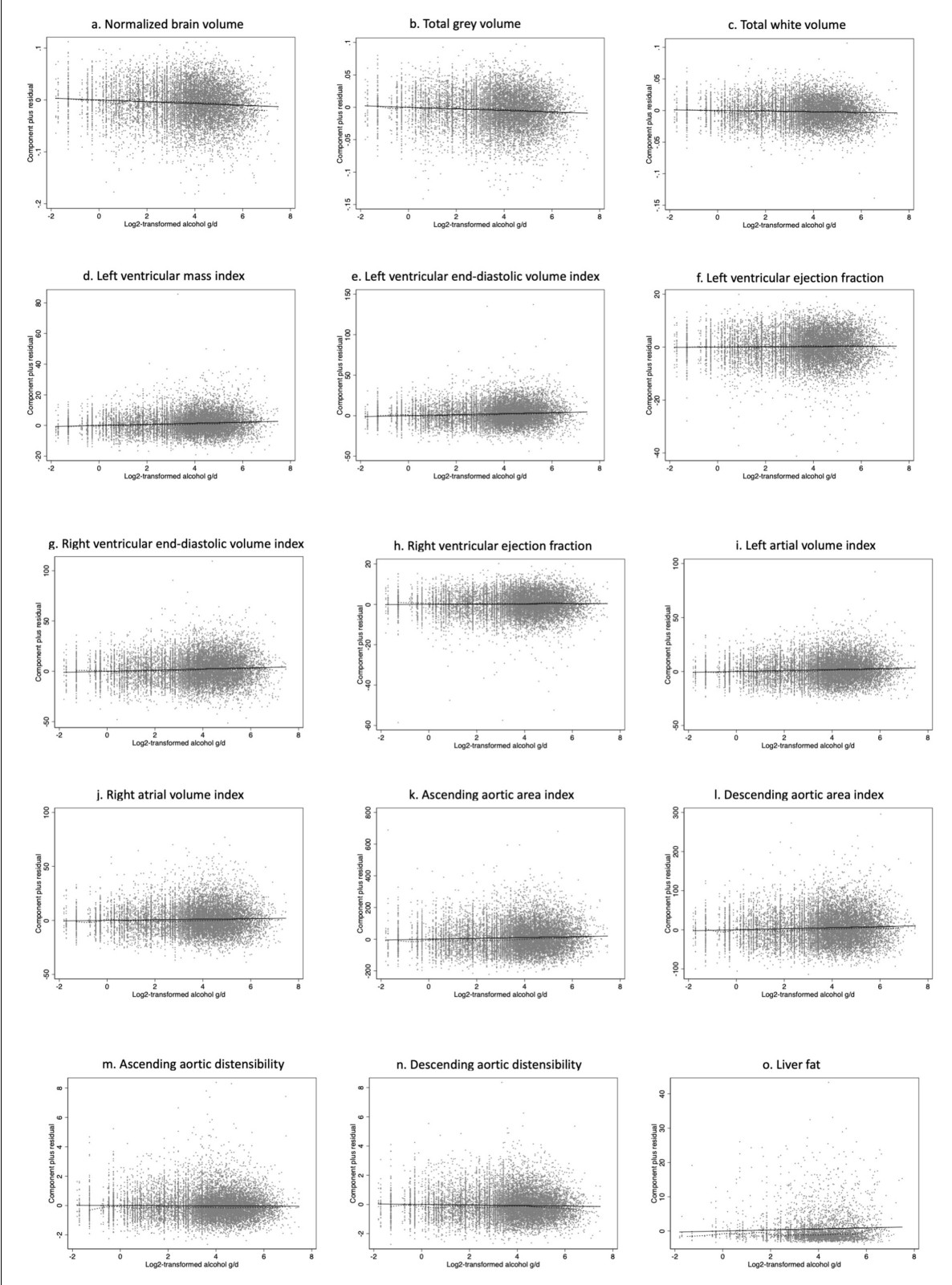

**Figure 2.** Partial residual plots for the imaging-derived phenotypes. Partial residual plots for (a) normalised brain volume, (b) total grey volume, (c) total white volume, (d) left ventricular mass index, (e) left ventricular end-diastolic volume index, (f) left ventricular ejection fraction, (g) right ventricular end-diastolic volume index, (h) right ventricular ejection fraction, (i) left atrial volume index, (j) right atrial volume index, (k) ascending aortic area index, (l) descending aortic area index, (m) ascending aortic distensibility, (n) descending aborting distensibility, and (o) liver fat.

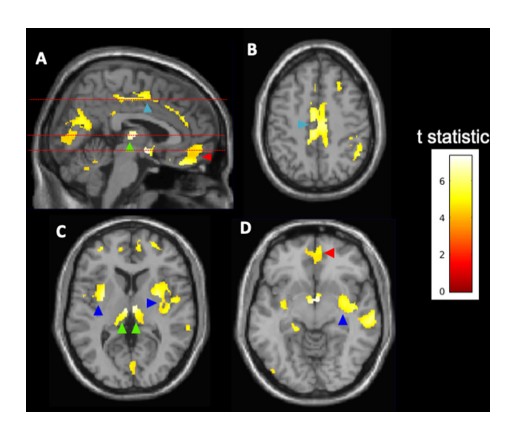

**Figure 3.** Voxel-wise associations of alcohol consumption with brain grey matter volumes (N = 10,143). Highlighted clusters define regions in which reductions of spatially normalised grey matter volume are inversely correlated with $log_2$-transformed alcohol consumption (g/d). The analysis suggests higher relative atrophy in the cingulate cortex (light blue arrowheads, **A, B**), thalamus (green arrowheads, **A, C**), orbital frontal cortex (red arrowheads, **A, D**), and insular cortex (dark blue arrowheads, **C, D**). Broken lines in (**A**) show levels for axial images in (**B–D**). The voxel-wise parametric model was adjusted for age, sex, ethnicity, body mass index, college degree education, hypertension, diabetes, and smoking history. The results are displayed on the MRI template available in SPM12 at axial slices of 46.5 mm (**B**), 6 mm (**C**), and –7.5 mm (**D**) relative to the bregma. The calibration bar provides the colour range use to describe t-scores calculated using a family-wise error (FWE)-corrected threshold of p<0.05.

range of alcohol intakes, indicating potentially pathological effects of alcohol on the brain, heart, and aorta across the full range of alcohol intake in the population, without evidence for a threshold.

Current guidelines for safe alcohol consumption vary between countries, mostly around one or two standard drinks/day. In the United Kingdom, the Chief Medical Officers' guideline for both men and women suggests that avoiding more than 14 units/week (corresponding to 16 g/d alcohol) on a regular basis maintains health risks at a low level (*UK Chief Medical Officer's, 2016*). In the United States, the suggested threshold is ≤2 drinks/day (~28 g/d) for men and ≤1 drink/day (~14 g/d) for women (*United States Department of Health and Human Services, 2015*). In a recent meta-analysis, excess mortality was observed above around 100 g alcohol intake per week (14.3 g/d), but with reduced incidence of myocardial infarction (*Wood et al., 2018*). Our results suggest that alcohol consumption below this threshold and below the currently recommended guidelines worldwide is associated with pathological structural and functional changes in brain, heart, aorta, and liver.

Previous research on structural and functional changes in the brain has indicated that excessive alcohol use is associated with abnormal development of the brain grey matter in animals (*Cosa et al., 2017*) and humans (*Heikkinen et al., 2017*), but the studies were small and underpowered. Here we found evidence of inverse associations of brain volume with amounts of alcohol consumed in a general population sample of adults. The magnitude of the effects appears to be meaningful: doubling of alcohol consumption (e.g., from 10 to 20 g/d) was associated with over half the brain volume reduction attributed to a year of aging in the population. This suggests a possible relationship between alcohol consumption and increased susceptibility to age-related brain pathologies and disease, consistent with our recent report of a genetic correlation between alcohol consumption and neuropsychiatric disease (*Evangelou et al., 2019*).

The effects of alcohol consumption on the brain appeared to be relatively generalised with a reduction in both white and grey matter volumes with greater alcohol consumption. Alcohol intake has previously been reported to be associated with reduced grey matter volumes in specific areas of the brain including the hippocampus and the inferior-medial frontal and anterior cingulate cortices (*Topiwala et al., 2017*). Our voxel wide-analysis also showed relatively greater associations of alcohol consumption with lower cingulate volumes and provides new evidence for similar directions of association within the orbital frontal cortex, the bilateral insula, and the thalami.

Studies of the effect of alcohol consumption on brain white matter have been less conclusive to date (*Ding et al., 2004*; *Mukamal et al., 2001*; *McEvoy et al., 2018*). This lack of consensus in the literature may in part reflect both study power and the potential complexity of changes in the macroscopic MRI measures of brain microstructure (*Ferizi et al., 2017*). Our study, the largest to date, found that the ratio of variance to effect was larger for white matter than grey matter, highlighting the need for larger study sizes to estimate volume differences in white matter. We also found higher fractional anisotropy and decreased orientation dispersion (a measure of greater fibre

**Table 3.** Associations of age and alcohol consumption with cardiac (N = 11,821) and aortic (N = 12,376) imaging phenotypes.

| | Aging/year, estimate* ± SE | p-value | Alcohol, estimate* ± SE | p-value |
|---|---|---|---|---|
| **Heart** | | | | |
| Left ventricular mass index | $-0.10 \pm 0.01$ | $1.3 \times 10^{-31}$ | $0.36 \pm 0.04$ | $8.5 \times 10^{-22}$ |
| Left ventricular end-diastolic volume index | $-0.33 \pm 0.02$ | $6.0 \times 10^{-90}$ | $0.61 \pm 0.07$ | $4.1 \times 10^{-17}$ |
| Left ventricular ejection fraction (%) | $0.04 \pm 0.01$ | $6.3 \times 10^{-7}$ | $0.05 \pm 0.03$ | 0.16 |
| Left atrial volume index (ml/m$^2$) | $-0.12 \pm 0.01$ | $8.9 \times 10^{-17}$ | $0.43 \pm 0.06$ | $6.0 \times 10^{-12}$ |
| Right ventricular end-diastolic volume index (ml/m$^2$) | $-0.33 \pm 0.02$ | $5.6 \times 10^{-80}$ | $0.57 \pm 0.08$ | $2.5 \times 10^{-13}$ |
| Right ventricular ejection fraction (%) | $0.04 \pm 0.01$ | $1.6 \times 10^{-8}$ | $0.05 \pm 0.03$ | 0.13 |
| Right atrial volume index (ml/m$^2$) | $0.04 \pm 0.02$ | $6.9 \times 10^{-3}$ | $0.26 \pm 0.07$ | $3.0 \times 10^{-4}$ |
| **Aorta** | | | | |
| Ascending aortic area index (mm$^2$/m$^2$) | $2.75 \pm 0.11$ | $4.0 \times 10^{-130}$ | $2.64 \pm 0.50$ | $1.5 \times 10^{-7}$ |
| Ascending aortic distensibility ($10^{-3}$mmHg$^{-1}$) | $-0.09 \pm 0.001$ | $<1 \times 10^{-300}$ | $-0.006 \pm 0.005$ | 0.22 |
| Descending aortic area index (mm$^2$/m$^2$) | $1.86 \pm 0.05$ | $2.0 \times 10^{-291}$ | $1.34 \pm 0.22$ | $2.1 \times 10^{-9}$ |
| Descending aortic distensibility ($10^{-3}$mmHg$^{-1}$) | $-0.09 \pm 0.001$ | $<1 \times 10^{-300}$ | $-0.02 \pm 0.005$ | $7.5 \times 10^{-4}$ |

*Estimates ± SE define coefficients for cardiac and aortic imaging phenotype changes per year age or per doubling of alcohol consumption (g/d) with their standard errors. The aging model was adjusted for sex, ethnicity, body mass index, and prevalence of college degree education, hypertension, diabetes, and smoking history. The alcohol consumption model was adjusted for age, sex, ethnicity, body mass index, and prevalence of college degree education, hypertension, diabetes, and smoking history. SE, standard error.

coherence) (*Enzinger et al., 2005*) in the corticospinal tracts with increasing alcohol consumption. As we have no reason to hypothesise adaptive plasticity with an increased density of descending motor neurons (one interpretation of increased fractional anisotropy) (*Mole et al., 2016*), we interpret these observations as reflecting not differences in the structures of corticospinal tracts, but lower densities in major axonal tracts *crossing* them (e.g., the cingulum bundle and superior longitudinal fasciculus). This interpretation is broadly consistent with the lower grey matter volumes in associated cortical regions (e.g., the cingulate and anterior prefrontal cortices) reported here and previously with greater alcohol consumption (*Topiwala et al., 2017*). A similar 'paradoxical' difference in fractional anisotropy in the context of decreased density of crossing fibre tracts distinguishes people with mild cognitive impairment who progress most rapidly to Alzheimer's disease (*Douaud et al., 2011*).

We provide evidence for cardiac remodelling including the association of alcohol consumption with larger ventricular masses, end-diastolic volumes, and left atrial volume indices. Our findings are consistent with previous echocardiographic studies that showed association between increasing alcohol intake and greater left ventricular mass (*Gonçalves et al., 2015a*; *Gémes et al., 2018*; *Manolio et al., 1991*) (although a smaller recent MRI study, assessing effects of light-to-moderate alcohol consumption, reported similar left ventricular mass in drinkers compared to non-drinkers; *Voskoboinik et al., 2019*). Left ventricular mass is a strong prognostic factor for incidence of cardiovascular disease and mortality (*Levy et al., 1990*). Greater atrial indices with higher alcohol consumption have also been reported (*Gonçalves et al., 2015a*); atrial enlargement can be considered a risk factor for several adverse cardiovascular outcomes (*Benjamin et al., 1995*) and is associated with increased risk of heart failure (*Gottdiener et al., 2006*). Alcohol consumption had same directions of effect as aging for the thoracic aortic measures (larger diameters and lower distensibilities). Such changes may also contribute to greater risks of cardiovascular disease and mortality (*Erbel and Eggebrecht, 2006*; *Redheuil et al., 2014*).

A number of studies have suggested a 'U'-shaped association between alcohol drinking and cardiovascular outcomes, even after exclusion of ex-drinkers from the non-drinker category (*Marmot et al., 1981*). Our diagnostic plots for the relationship between alcohol and cardiovascular measures did not provide evidence of deviation from a monotonic association and therefore did not support a 'U'-shaped association for these measures. Rather, our results point to pathological effects of regular alcohol intake on the heart and major vessels occurring below current consumption guidelines.

Excessive alcohol intake is a well-known risk factor for increased liver fat (*Bellentani et al., 1997*), but the evidence regarding low-to-moderate alcohol consumption has been inconclusive. Several prospective studies have reported a lower prevalence or risk of fatty liver (*Moriya et al., 2015*; *Hashimoto et al., 2015*; *Yamada et al., 2010*) for low-to-moderate compared to excessive alcohol consumption. However, a randomised trial showed that even moderate consumption of red wine for three months increases liver fat (*Kechagias et al., 2011*; *van Eekelen et al., 2019*). Our results are consistent with this. The magnitude of the association with alcohol (0.15% per doubling of alcohol consumption) suggests that alcohol could explain a major proportion of the population variance in liver fat (mean liver fat density percentages in a larger UK Biobank sample ranged between a mean of 1.34–5.71%) (*Linge et al., 2019*).

Our results highlight the multi-organ effects of low and moderate levels of alcohol consumption in late middle-aged people. This may reflect direct toxicities. For example, the toxic effect of alcohol or its metabolites can cause myocardial damage that leads to increased rates of cardiomyopathies, heart failure, and mortality (*Wood et al., 2018*; *Gonçalves et al., 2015b*). Deleterious effects of alcohol on cell function and survival and consequent organ injury may affect a number of biological processes, e.g., oxidative stress, inflammation, aberrant post-translational modifications of proteins, dysregulation in lipid metabolism, upregulation of catabolic processes and signal transduction pathways, and epigenetic changes involving DNA methylation impairments (*Souza-Smith et al., 2016*; *Osna and Kharbanda, 2016*). However, our observations may also reflect interactions between pathologies across organ systems. For example, alcohol is metabolised primarily in the liver, where it increases fatty acid synthesis (*Cederbaum, 2012*) and leads to metabolic abnormalities that independently contribute to cardiovascular and brain pathologies (*Suzuki et al., 2019*; *Wilson et al., 2005*; *Cai et al., 2012*).

Our study has limitations. The observational design of UK Biobank has inherent limitations that preclude establishing causal relationships. Also, the participants are relative healthy compared to the general UK population and most are of European ancestry (*Fry et al., 2017*). The extent to which results are generalisable to other populations or populations of various ethnic groups needs to be explored. We relied on self-reported alcohol intake information obtained on a single occasion at the baseline assessment. This is subject to misreporting and recall bias, especially among heavy drinkers who may under-report their intake (*Boniface et al., 2014*; *Greenfield and Kerr, 2008*) and may have led to bias in the estimation of the effects of alcohol intake on the outcomes under study. Furthermore, the imaging data were also only available at one point in time. Future longitudinal analysis will enable effects of low or moderate alcohol intake on anatomical and structural changes of the various organs to be measured over time directly.

In summary, our findings provide new insights into the adverse effects of alcohol intake on the structures of brain, heart, and aorta and on liver fat deposition. Specifically, we show that these effects are monotonic (linear-log), increasing across the full range of reported alcohol intakes in this large population sample, with no apparent threshold. Our results therefore indicate that pathological changes in major organ systems may occur at even small amounts of daily alcohol intake. This has important implications for governmental and public health policies concerning 'safe' levels of alcohol drinking in the general population. Further research is needed, but we believe current guidelines on alcohol drinking may need to be revisited.

Disclosures EE acknowledges consultancy fees from OpenDNA. PMM acknowledges consultancy fees from Roche, Adelphi Communications, Celgene, and Biogen. He has received honoraria or speakers' honoraria from Novartis, Biogen, Medscape, Adelphi Communications, and Roche and has received research or educational funds from Biogen, Novartis, GlaxoSmithKline, and Nodthera. PE is a member of the UK Biobank Strategic Oversight Committee (previously the UK Biobank Steering Committee).

## Acknowledgements

PMM gratefully acknowledges generous support from Edmond J Safra Foundation and Lily Safra, the NIHR Investigator programme, and the UK Dementia Research Institute, which is supported by the MRC, the Alzheimer's Society, and Alzheimer's Research UK. PMM also acknowledges support from the National Institute for Health Research (NIHR) Imperial Biomedical Research Centre. HS is supported by the Grants-in-Aid program from the Japan Society for the Promotion of Science

(20K07776). RP holds a fellowship supported by Rutherford Fund from Medical Research Council (MR/R0265051/1 and MR/R0265051/2). PE is director of the Medical Research Council (MRC) Centre for Environment and Health and acknowledges support from the MRC (MR/L01341X/1; MR/S019669/1). PE also acknowledges support from the National Institute for Health Research (NIHR) Imperial Biomedical Research Centre and the Imperial College British Heart Foundation Centre for Research Excellence (RE/18/4/34215). He is a UK Dementia Research Institute (DRI) Professor, UK DRI at Imperial College London (MC_PC_17114). PE is associate director of Health Data Research UK for London, which is supported, among others, by MRC, NIHR, Engineering and Physical Sciences Research Council, Economic and Social Research Council, Wellcome Trust, and British Heart Foundation.

## Additional information

### Competing interests

Evangelos Evangelou: E.E. acknowledges consultancy fees from OpenDNA. Paul M Matthews: PM acknowledges consultancy fees from Roche, Adelphi Communications, Celgene and Biogen. He has received honoraria or speakers' honoraria from Novartis, Biogen, Medscape, Adelphi Communications and Roche and has received research or educational funds from Biogen, Novartis, GlaxoSmithKline and Nodthera. The other authors declare that no competing interests exist.

### Funding

| Funder | Grant reference number | Author |
|---|---|---|
| Medical Research Council | MR/R0265051/1 | Raha Pazoki |
| Medical Research Council | MR/R0265051/2 | Raha Pazoki |
| Medical Research Council | MR/L01341X/1 | Paul Elliott |
| British Heart Foundation | RE/18/4/34215 | Paul Elliott |
| Medical Research Council | MR/S019669/1 | Paul Elliott |
| Japan Society for the Promotion of Science | 20K07776 | Hideaki Suzuki |

The funders had no role in study design, data collection and interpretation, or the decision to submit the work for publication.

### Author contributions

Evangelos Evangelou, Conceptualization, Data curation, Formal analysis, Methodology, Writing - original draft, Writing - review and editing; Hideaki Suzuki, Wenjia Bai, Raha Pazoki, Formal analysis, Writing - review and editing; He Gao, Data curation, Writing - review and editing; Paul M Matthews, Paul Elliott, Conceptualization, Data curation, Supervision, Methodology, Writing - original draft, Writing - review and editing

### Author ORCIDs

Evangelos Evangelou  http://orcid.org/0000-0002-5488-2999
Raha Pazoki  https://orcid.org/0000-0002-5142-2348
Paul M Matthews  https://orcid.org/0000-0002-1619-8328
Paul Elliott  https://orcid.org/0000-0002-7511-5684

### Ethics

Human subjects: In UK Biobank, ethical approval for data collection was received from the North-West Multi-centre Research Ethics Committee (REC reference: 11/NW/0382) and the research was carried out in accordance with the Declaration of Helsinki of the World Medical Association. No additional ethical approval was required for the analyses of the data.

Decision letter and Author response
Decision letter https://doi.org/10.7554/eLife.65325.sa1

## Additional files

### Supplementary files

• Supplementary file 1. Alcohol consumption (g/d) in all and males and females for brain grey matter (N = 10,143), brain white matter (N = 9053), heart (N = 11,821), aorta (N = 12,376), and liver (N = 3649).

• Supplementary file 2. Coefficients for log2 alcohol and age in an expanded model for cardiac (N = 11,821) and aortic (N = 12,376) imaging phenotypes including tests of interactions between log2 alcohol and age.

• Transparent reporting form

• Reporting standard 1. STROBE checklist.

### Data availability

For this project, UK Biobank has granted access to our team through approved applications with ID #13375 and #18545. Data is available to bona fide researchers through application to UK Biobank as described here: https://www.ukbiobank.ac.uk/enable-your-research/apply-for-access. Guidance on how to apply for the various types of UK Biobank data can also be found at this link.

The following dataset was generated:

| Author(s) | Year | Dataset title | Dataset URL | Database and Identifier |
|---|---|---|---|---|
| UK Biobank | 2021 | UK Biobank data showcase | https://biobank.ndph.ox.ac.uk/ukb/ | UK Biobank, biobank.ndph.ox.ac.uk/ukb/ |

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
