## [Decision Letter]

**Acceptance summary:**

The authors assessed global effect sizes of alcohol consumption on quantitative MRI measures in the brain, heart, aorta and liver using a large sample of UK biobank data

The results suggest that for both genders alcohol consumption below current recommended thresholds in guidelines world wide is associated with pathological structural and functional changes in brain, heart , aorta and liver. The findings are of major interest to those conducting research into these tissues, those conducting research on alcohol consumption and health and to policy makers.

**Decision letter after peer review:**

Congratulations, we are pleased to inform you that your article, "Alcohol consumption in the general population is associated with structural changes in multiple organ systems", has been accepted for publication in *eLife*. Your manuscript has been reviewed by 3 reviewers, one of whom is a member of our Board of Reviewing Editors, and the evaluation has been overseen by a Senior Editor.

Suggest modifying the sentences on lines 142 and 143 with addition of the word "ventricular" before left and right end-diastolic and end-systolic volumes. Also, add "left ventricular" end-systolic volume. Otherwise no need to modify text.

*Reviewer #1:*

The authors have studied a large sample of UK biobank data with a view to getting a feel on what is the safe alcohol intake with no harmful effect on human organs.

One of the major strength is the large sample size and meticulous measurement and quantification of organ involvement including brain, heart and aorta.

One of the weakness is the self reported alcohol intake which may not be necessarily true on many occasions.

However the findings are useful and alarming as it seems to be that there is no safe alcohol limit without any organ involvement as the authors reported here.

This make sense given the toxic effects of alcohol in multiple biological systems in the human body.

Prospective studies with accurate alcohol measurements and further biological underpinning on how these organ involvement occur is worthy of exploring.

This study is very well done and findings are worthy of publishing as it is.

*Reviewer #2:*

Excessive alcohol consumption has multiple well known adverse effects but multiorgan effects across much lower and usual ranges of consumption are less well defined and the current interpretation of the available data has been translated into safe thresholds for public health purposes.

The authors set out to assess global effect sizes of alcohol consumption on quantitative MRI measures in the brain, heart, aorta and liver. Their subjects are a subset of the half a million very well characterized participants in the UK Biobank study for whom brain, cardiac, aortic and liver MRI and image derived phenotypes were available. There were appropriate exclusions including non drinkers and those with established disease in these organs. The alcohol intake was self reported from a touch screen questionnaire and was converted into grams of alcohol per day. Alcohol intake was log transformed so that a change of 1 unit translates into a doubling of alcohol consumption.

The MRI methodology is described in detail. The age related changes are factored in and alcohol adds to further adverse changes There are adverse effects on all the tissues examined with doubling of alcohol intake with no lower safe limit There is reduced brain grey and white matter with greatest adverse effects in the cingulate and orbital frontal cortices, the bilateral insula and thalami; changes in left ventricular and atrial mass and volume, aortic changes and increased liver fat.

The results suggest that for both genders alcohol consumption below current recommended thresholds in guidelines world wide is associated with pathological structural and functional changes in brain, heart , aorta and liver.

Major strengths are the large size of the study, the well characterized subjects and the detailed MRIs coupled with a very clear presentation of the study and its implications.

There are no obvious weaknesses however limitations as well discussed by the authors are (1). that the study is cross sectional precluding establishment of a causal relationship which requires further longitudinal studies. (2). that the alcohol intake is self reported and heavy drinkers may under report intake. (3). that the population studied is mainly Caucasian.

The authors have achieved their aims and the results support their conclusions.

The findings are of major interest to those conducting research into these tissues, those conducting research on alcohol consumption and health and to policy makers. They provide ample scope for debate between those who will want to reduce the currently existing thresholds further and those who have the opposite view and will argue that this is preliminary data lacking causal effects.

A very well executed and presented study of great interest. I don't see any need to modify the text.

*Reviewer #3:*

Suggest modifying the sentences on lines 142 and 143 with addition of the word "ventricular" before left and right end-diastolic and end-systolic volumes. Also, add "left ventricular" end-systolic volume.